# Optimizing Entomopathogenic Nematode Genetics and Applications for the Integrated Management of Horticultural Pests

**Mahfouz M. M. Abd-Elgawad** 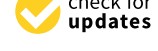

Plant Pathology Department, Agricultural and Biological Research Institute, National Research Centre, El-Behooth St., Dokki, Giza 12622, Egypt; mahfouzian2000@yahoo.com

**Abstract:** Entomopathogenic nematodes (EPNs) can kill and recycle in their host populations, which bodes well for EPNs' exploitation in long-term and safe pest management. However, EPNs' cost and efficacy need transformational technology to supplant less expensive and more effective but toxic/unhealthy pesticides. A technology that allows for the significant uptake of commercial EPNs should both boost their market suitability and provide genetic improvements. This review provides brief overviews of EPNs' biology and ecology from the standpoint of pest/pathogen management as a prerequisite for EPN improvements. Understanding the biology and ecology of EPNs, particularly their symbiotic relationships with bacteria, is crucial to their effective use in pest management. This review provides relevant insights into EPN-symbiotic bacteria and the EPN–symbiont complex. The symbiotic relationship between EPNs and bacteria plays a key role in IPM, providing unique advantages. Either of them can be included in mechanisms underlying the various positive sides of plant–insect interactions in emerging integrated pest management (IPM) systems. Recent approaches, in which EPNs can act additively or synergistically with other production inputs in IPM programs, are discussed for further expansion. The simultaneous favorable effects of EPNs and/or their mutualistic bacteria on several pest/pathogen species of crops should be identified. Merits, such as the rapid killing of insect pests, ease of EPN/the symbiont's mass production and a broad host range, are presented in order to widely disseminate the conditions under which EPN usage can offer a cost-effective and/or value-added technique for IPM. To maximize the effectiveness of EPNs in IPM, various genetic improvement techniques are being explored. Such techniques, along with their merits/demerits and related tools, are reviewed to optimize the common biocontrol usage of EPNs. Examples of genetic improvements to EPNs that allow for their use in transformational technology, such as a cost-effective application technique, increased infectivity, and toleration of unfavorable settings, are given. Proper production practices and genetic techniques should be applied carefully to avoid undesirable results; it is suggested that these are considered on a case-by-case basis. This will enable us to optimize EPN performance based on the given variables.

**Keywords:** biocontrol; entomopathogenic nematode; plant-parasitic nematode; marketing



## 1. Introduction

Entomopathogenic nematodes (EPNs) in the two main genera *Heterorhabditis* and *Steinernema* and their bacterial symbionts [1] form a notable ground for their practical and further use as biocontrol agents (BCAs) of many insect pests. The two genera relate to two extensionally convergent families, Heterorhabditidae and Steinernematidae, which are consolidated via their ordinary use of symbiotic bacteria of the genera *Photorhabdus* and *Xenorhabdus*, respectively. The bacteria are crucial to the biocontrol of their insect hosts. Clearly, the two families have developmental tactics to successfully achieve this useful task, along with their nutritional linkage to these bacteria. The role and implications of this mutualism are exceptional. Contrary to other entomophilic nematode groups for

which mass propagation is difficult, the symbionts of EPNs can easily turn numerous proteins into an optimized diet for the efficient development and multiplication of their nematode partners. Thus, their ability to readily provide huge EPN numbers via various in vitro culture techniques has caused them to rapidly progress from mostly unknown entomopathogens to widely researched and market-available BCAs.

Due to their high safety profile concerning human beings, non-target organisms and the environment [2,3], EPNs are broadly excepted from pesticide registration demands in many states. This merit has contributed to EPNs' commercialization. Recently, the commercial development of at least five *Heterorhabditis* species and eight *Steinernema* species was reported [4]. Examples include *H. bacteriophora*, *H. indica*, *S. carpocapsae*, *S. feltiae* and *S. riobrave*. Moreover, cottage industries can generate additional species to biocontrol targeted pest species on demand. The overall framework of EPN experimentation and its advantages has been introduced by relevant books/articles [5–11]. Alongside EPNs' merits, there are a few hurdles to their further commercial usage for integrated pest management (IPM). Their comparatively high price and unstable efficacy are the two major barriers. Hence, programs for EPN improvement are ongoing based on two substantial approaches, i.e., enhancing EPNs' effectiveness and increasing their commercial competence. This competence is being challenged by increasing EPNs' production efficiency and consistency as well as their adequate formulations and application technology. Although specific procedures to achieve the objectives necessary for this competence have previously been reviewed [10,11], new horizons to expand EPNs' utility against additional pests of economic significance, as well as the related activity of their mutualistic bacteria individually, are highlighted herein. As this review focuses on enhancing EPNs' effectiveness via genetic improvement plans, a brief review of their biology and ecology, as well as their application methods, are presented as a background to the current knowledge base.

## 2. EPN Biology and Ecology

Due to the continuous and global nematode surveys, the species of EPN and their mutualistic bacteria are being elevated in great numbers. Currently, about 22 *Heterorhabditis* and 102 *Steinernema* species have been recorded [11–13]. EPNs have been isolated from all continents except Antarctica. Their mutualistic bacteria are reported as including 20 *Photorhabdus* [14] and 27 *Xenorhabdus* species [15]. The only free living EPN stage is the specialized third-stage juvenile, called the infective juvenile (IJ) or dauer juvenile. This can actively seek and invade the host insect. The foraging strategies of these juveniles occupy two distinct ends of the ambusher–cruiser continuum. However, their foraging practices are usually constricted by various factors, e.g., soil texture/properties, signals from the host insects/plant roots being parasitized by these arthropod hosts, and volatile cues, as recently reviewed [16]. IJs can release cues that negatively affect the activity of root insect herbivores [17]. This fact may increase the benefits of utilizing EPNs in IPM strategies.

EPN invasion into an arthropod host may occur through the host's natural openings (spiracles, mouth, and anus) or by directly boring through the attacked insect at parts with a thin cuticle [7]. Once it has entered the hemocoel, the sophisticated link between EPN and the symbiont starts to function within the insect host, releasing mutualistic bacteria via EPN regurgitation or defecation. The bacteria routinely induce insect mortality via septicemia/toxemia. This is a common activity of the natural *Xenorhabdus–Steinernema* or *Photorhabdus–Heterorhabditis* complex [14,15]. This insect mortality in a susceptible host is realized within a relatively short time (ranging from a few hours to about 3 days). *Steinernema* spp. IJs develop into adults (females and males) and multiply several times within the host to produce both males and females (except for *Steinernema hermaphroditum*) [4]. *Heterorhabditis* spp. IJs grow into hermaphroditic adults in the first generation. Then, their following generations contain hermaphroditic individuals as well as both males and females. Within the infected insect cadaver, the EPNs reproduce by feeding on the host tissues decomposed by the bacteria and the bacteria themselves. They can complete several generations before the nutrient resources of the host cadaver are exhausted within, at most,

4 weeks [4]. Interestingly, thousands of IJs containing the bacteria in their digestive system depart from the insect cadaver in search of other insect hosts. These mutualistic bacteria rely on their partners, nematodes, to infect a new host. They can create antibiotics that disable secondary host infestations while setting an adequate diet for the EPN feeding inside the cadaver body. The definitive role of the nematodes and symbionts in beating the immune response of their hosts has been examined in some nematode–bacteria complexes for certain insect species. Recent studies proved that EPNs are also significant partners in inducing insect mortalities [18,19].

## 3. Potential of the Symbionts in IPM

Former EPN-relevant research relied mainly on the notion that the mutualistic bacteria cannot actively remain, with infective capabilities, outside their nematode partners or host insects. Nonetheless, recent reviews showed that the mutualistic bacteria of *Heterorhabditis* spp., i.e., *Photorhabdus* spp. [14], and *Steinernema* spp., i.e., *Xenorhabdus* spp. [15], can individually be contained in mechanisms underlying the favorable aspect of plant–pest or –pathogen interactions, especially for modern cropping systems. In this respect, the mode of action of the species related to these two bacterial genera may involve either genetic manipulation or direct antimicrobial activity. Thus, they could be comprised in the biocontrol of plant–insect pests not only via transgenic plants but also through other application methods, including spraying, alginate beads, dipping in suspension, pellet, powder, topical application and/or synergism with other biopesticides [14,15]. Moreover, they could be applied against various pests, such as plants, animals and medical insects, and mites as well as pathogens, e.g., fungi, bacteria, plant-parasitic nematodes, arboviruses, oomycetes and protozoa [10,14,15]. Such applications do not negate the fundamental role of these bacteria for commercial or in vitro mass production of EPNs. Both individual and combined bacterial species of *Photorhabdus* and *Xenorhabdus* should be significantly used to manage plant pathogens/pests through IPM plans. Details for fixing *Xenorhabdus*- and *Photorhabdus*-obtained insecticidal, acaricidal, antimicrobial, fungicidal, pharmaceutical, nematocidal and toxic compounds into present and emerging IPM strategies to control numerous pests/pathogens has been the recent focus of attention [14,15,20]. For example, a product called 'Col-Kill' containing *Bacillus thuringiensis tenebrionis* with the *Photorhabdus temperate temperata* culture broth proved its effectiveness against *Phaedon brassicae* (Coleoptera: Chrysomelidae), the brassica leaf beetle [21]. Another product named 'Dual Bt-Plus' consists of a *Xenorhabdus nematophila* culture broth with *B. thuringiensis aizawai* and *B. thuringiensis kurstaki* and has demonstrated efficacy in controlling two lepidopteran pests, i.e., *Plutella xylostella* (the diamondback moth) and *Spodoptera exigua* (the beet army worm) [22]. Moreover, da Silva et al. [20] reviewed some species of *Photorhabdus* and *Xenorhabdus* with their metabolites that can enhance their pesticidal potential against definite insect pests, e.g., *P. luminescens* mixed with *B. thuringiensis kurstaki* suppresses the development of the cotton leafworm. Those authors attributed the merits of these mutualistic bacteria to their genetic constitution, which can encode low-molecular-weight secondary metabolites/toxins with insecticidal, antifungal, antibiotic and antiparasitic activities. Eventually, these bacteria will have an excellent arsenal for IPM of many economically important crops.

Notwithstanding the evidenced merits of these bacteria, the full potential of their biocontrol efficacy have yet to be realized. The relevant utilization of *Xenorhabdus* spp. has not been promptly advancing, especially due to costs and problems linked to their commercial production. On the contrary, *Photorhabdus* spp. have proved to be relatively more suitable because of a recent breakthrough in the cheap mass production of *Photorhabdus* spp. [23]. Additionally, *Photorhabdus* individually can be used for relatively more goals and against more plant pathogens/pests than *Xenorhabdus* [14,24]. On the other hand, as the prevailing setting of living for species of both genera (*Xenorhabdus* and *Photorhabdus*) and their mutualism with relevant EPN-IJs, they have a common issue too. It is related to the natural survival of these bacteria within an uneven distribution of their nematode

partners [25]. Nevertheless, current progress in mastering the genome sequencing that reflects the hidden arsenal of *Photorhabdus* spp. [26] and *Xenorhabdus* spp. [27] may facilitate their manipulation in IPM. The strong pathogenic power of *Photorhabdus* and *Xenorhabdus* spp. against a wide range of arthropods, as well as their reliability in controlling specific pests/pathogens and versatility are clear signs of the feasibility of integrating them into novel management strategies as a way forward in combating pests/pathogens and crop protection [28–33]. Examples are presented (Figure 1) to show these EPN–symbiont complexes killing and multiplying in their insect hosts and then leaving them to search for other insect hosts.

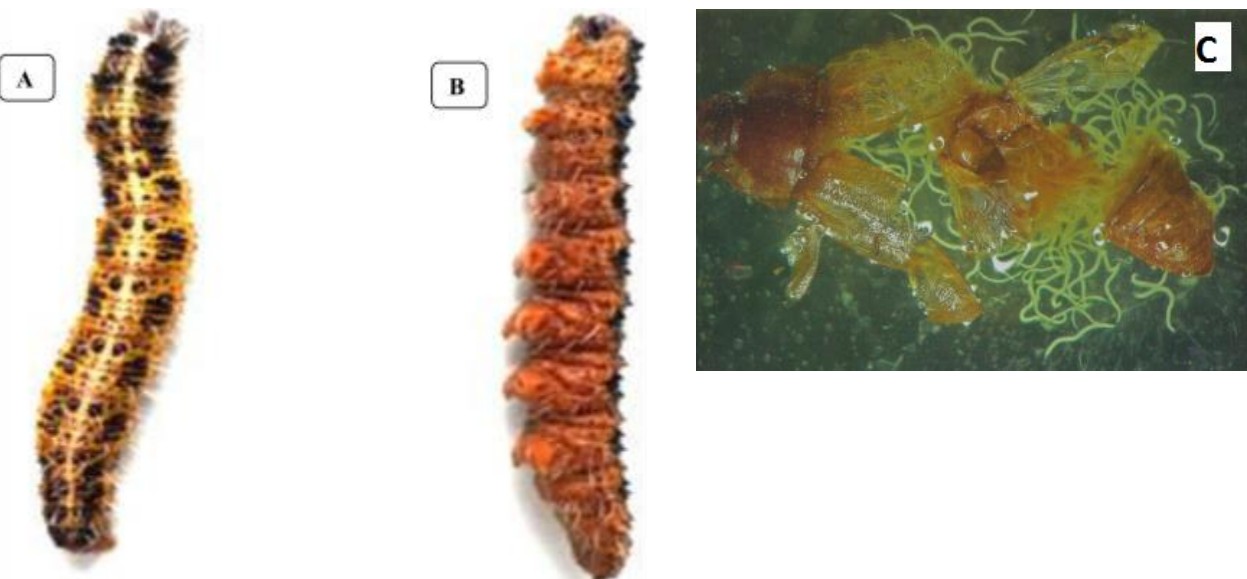

**Figure 1.** Dead larvae of cabbage butterfly, *Pieris brassicae* (**A**) killed by *Steinernema* sp., (**B**) killed by *Heterorhabditis* sp., and (**C**) emergence of *H. bacteriophora* after parasitizing an adult of the strawberry sap beetle *Lobiopa insularis* [33].

## 4. Potential of the EPNs in IPM

### 4.1. General Precautions for Optimal EPN Applications

Perfect EPN–host matching (i.e., selecting the ideal nematode species or strain) is imperative to realize the optimal effect in the targeted pest management zone. Also, the ecological surroundings must be determined. Thus, the characteristics of the EPN isolate/strain/species that can lead to the best control of a particular insect species/strain under given ecological and biological sets represent the perfect matching. An EPN that effectively controls a specific white grub that feeds on turfgrass may show far less efficacy on another grub species of the same area/habitat, or even on another instar/stage of the same insect species [34]. The persistence of the selected nematode strain or species should also be considered. It reflects the capacity of nematode reproduction within their target hosts and the period of IJ survival in the given environment. Numerous factors that can modulate such characteristics have recently been reviewed and should be considered [4,16].

Other basic precautions and practices must be carefully deemed [16]. The EPN application must be introduced away from UV radiation to avoid its killing effect on the IJs, preferably before sunrise or at sunset. A plausible approach to circumvent the UV damage to IJs is to use UV protectants mixed with the IJ suspension. An alternative method is to apply soil IJs in high carrier volumes immediately followed by rinsing with additional reasonable amounts of irrigation in order to lessen EPN misplacements. The optimal temperature activity range of the used EPN should be employed. Most EPN strains have their best biocontrol efficacy within 20–30 °C. Yet, few EPN strains are vigorous at low temperature of cold areas from where they were isolated and vice versa for others, i.e., they

thrive in relatively hot semiarid ranges. These uncommon nematode strains offer promise for their use in their original environments [16].

Numerous organisms that live in the soil or on its surface represent biotic agents harmful to EPNs for they have a living stage (IJs) in the soil or within their arthropod hosts. These unfavorable biotic factors may feed on the nematodes and/or their hosts. Moreover, soil-borne pathogens may either synergize or compete with EPNs present in the same environment for shared hosts [4].

*4.2. Applications of EPNs in IPM*

Initially, both favorable and unfavorable interactions between EPNs and other biotic and edaphic/abiotic factors should be manipulated to optimize the biocontrol activities of EPNs, especially via IPM schemes [30–32]. The referenced reviews provided specific combinations of BCAs that could boost IPM programs and courses that make them fit for value-added, cost-effective techniques. The EPNs and entomopathogenic fungus (EPF) could achieve significant synergism when usage of the fungus *Metarhizium anisopliae* is followed within 7–14 days by EPNs. These interactions resulted in up to 100% mortality of the third instar of *Otiorhynchus sulcatus* (black vine weevil) larvae. The authors in [35] concluded that these combined effects may offer not only reliable management or full mortality of the pest but also economically feasible techniques for *O. sulcatus* larval control. Also, applying *Amblyseius cucumeris* (a predatory mite) combined with *H. indica* or *H. bacteriophora* attained up to 83% suppression in *Frankliniella occidentalis* (the flower thrips) populations [36]. Moreover, populations of the latter pest found in the glasshouse were reduced by up to 74% when treated with *S. carpocapsae* combined with either the insecticidal fungus *M. anisopliae* or a neem extract formulation [37].

On the other hand, Koppenhöfer's work team at Rutgers University (USA) theorizes that synergistic effects may also stem from combining chemical insecticides with EPNs. The team accomplished such synergisms and consequently advocated that such combinations could control noxious pests and minimize both the cost of EPN usage and chemical pesticide frequency/dosage [34,38]. Hence, detecting new EPN species/strains that are particularly reliable for scarab control, such as *Steinernema scarabaei*, may offer continuous benefits and widen these approaches [32,34].

Likewise, the dual-purpose usage of EPNs seems promising against more than one crop pest related to various groups. This is mainly due to the general wide range of insect host species and/or other pests that can be managed by EPNs. Three EPN populations could significantly ($p < 0.05$) suppress populations of the root-knot nematode (RKN) *Meloidogyne incognita* (a serious plant-parasitic nematode (PPN) species) on watermelon plants. Abd-Elgawad [30] speculated that in such cases, EPNs would control not only susceptible insect pests commonly associated with the plant roots, such as *S. littoralis*, but also *M. incognita*. Moreover, many economically important vegetables such as tomato [39], pepper [40] and potato [41] may suffer severe yield losses due to growing in RKN-infested fields. Such fields are frequently infected with several insect pests simultaneously, mostly those belonging to the order Lepidoptera and are highly susceptible to EPNs [30] or their symbionts [14,15]. As applications of these BCAs can suppress infection by PPNs and reduce populations of insect pests, what is usually measured against individual pests/pathogens could sometimes be an underestimate of the EPN/symbiont application value [42]. This is especially important because considerable damage caused by lepidopteran pests and RKNs affect other chemical pesticides-sensitive crops, such as berseem clover and other forage legumes [43]. Because of the nature of these forages as feeding diets for livestock, restrictions have been imposed on the use of such toxic chemicals. Besides decreasing populations of insect pests, the application of the mutualistic bacteria can suppress infections by plant pathogenic fungi [44]. Hence, wise application of EPN/symbiont may alleviate multiple pathogen/pest problems via keeping their populations below the economic threshold level. Likewise, two insect pests, *Otiorhynchus ovatus*

(strawberry root weevil) and *O. sulcatus*, on strawberry plants could be well controlled by EPNs with a decrease of 75% or more in their populations on treated plots [45,46].

Such approaches where EPNs can function collectively or additively/synergistically with other production inputs in IPM programs should be expanded to include arable crop production and low-value crops [31,32]. This would circumvent the higher expenses of EPN costs relative to chemical insecticides or other biological formulations, such as *Bacillus thuringiensis*. Currently, relatively high EPN costs impede their use on major-acreage row crops in most countries. This does not negate the fact that earnest and successful attempts could use EPNs on some arable crops, such as alfalfa (*Medicago sativa*), to manage *Otiorhynchus ligustici* (alfalfa snout weevil). Optimistically, researchers from Cornell University (USA) have developed strategies whereby farmers in New York can recover, culture and apply their own nematodes to effectively control this naturalized alfalfa pest. Interestingly, the introduced EPNs can persist within alfalfa fields for multiple years across crop changes [47,48]. The latter author [48] reported that applying *S. carpocapsae* formulated with *S. feltiae* could bring *O. sulcatus* populations also in fields planted with strawberry and cranberry down to sub-economic threshold levels from their outbreak conditions.

Successful uses of EPNs in IPM in these examples have typically adopted optimal application strategies. These comprised the appropriate compatibility/tolerance of EPNs to the existing abiotic factors, especially temperature, desiccation and ultraviolet light, as well as biotic factors, e.g., EPN dose and virulence, EPN-host matching and resistance to nematicides. Thus, it is imperative to define and avoid unfavorable factors. For instance, *Synanthedon bibionipennis* (Strawberry crown moth) is so susceptible to *S. carpocapsae* and *H. bacteriophora* that the two EPN species achieved almost full insect mortality in the laboratory bioassays. Yet, applying these EPNs to *S. bibionipennis*-infected strawberry fields led to much lower *H. bacteriophora* and *S. carpocapsae* efficacy (just 33% and 51% infection rates, respectively) against this main root-boring Sesiid pest of strawberry plants when applied during late fall [49]. The authors attributed such low rates to the impact of abiotic factors, mostly temperature, which lessened effective *S. bibionipennis* control.

Ultimately, as with other BCAs [50], optimizing safe pest control and boosting crop productions via the sequential, dual-purpose and co-application of EPNs with other agricultural inputs should be earnestly attempted and considered on a wider basis too. Because these techniques can offer improved crop yield and protection strategies via continuous fine-tuning, research priorities for wise incorporation of EPNs in sustainable agricultural regimes should always be updated. The potential that more EPN species/strains are being identified or are likely to be widely available soon [4,51,52] beckons to be further exploited for IPM plans.

## 5. Genetic Techniques to Enhance EPN Efficacy

As previously mentioned, EPN improvement still requires more developments in commercial competence or suitability. Hence, not only do EPN combinations with other agricultural inputs entail further applications but also boosting EPN mass production, strain development, formulation/application technology, expansion of their target pest/pathogen species, and careful manipulation of relevant biotic/abiotic factors should be moved forward to broaden reliable and inexpensive EPN uses [4,53,54]. Traditional and modern genetic improvement programs have been basic to commercial competition up to now. The programs aim at enhancing nematode infectivity to specific target pests and/or boosting EPN tolerance to unfavorable ecological conditions affecting their biocontrol activity. They start with efficient techniques of sampling and extraction to logically expand the nematodes' genetic pool and/or discover promising isolates. A functional sampling technique in order to obtain high nematode recovery as well as elevated frequency values relative to other techniques was suggested [55]. The technique employs four combined factors. These factors, for the high hit rate (61.7%) of EPN extraction, include favorable method of sampling, fitting of both location and timing, and using multiple extraction procedures.

Also, a thorough analysis of EPN production methods with their merits/demerits for relevant exploitation and favorable formulations were recently summarized [56]. Three production methods have significantly contributed to the introduction of EPNs to the marketplace globally. The fine-tuning of these methods to reduce costs and increase EPN virulence should continue to contribute to EPNs expanded applications. In vitro liquid mass production is the best economical method and is likely ongoing to maintain the quantity of global EPN production. On the other hand, in vitro solid production may still be competitive, especially where labor is less costly. Although in vivo production is the least economical method, it will likely become adequate for certain niche markets or for certain small or start-up firms. Further optimization of in vivo production [30–32] may boost cost efficiency. Meanwhile, it is likely that the economy of scale is reliable for boosting in vitro approaches, including both methods of solid and liquid media [56]. For example, pheromone extracts that act as boosters for EPN efficacy were reported [57] and may be promising components in the media of these in vitro approaches. As genetic improvements of EPNs should continue to upgrade EPNs pest control capabilities, the following part initially addresses various features of insect–EPN interactions.

### 5.1. General Aspects of Insect-EPN Interactions

Because EPNs and their host insects coevolve in nature, their relevant genes have supposedly balanced co-existence for encoding via the attacking and defending interactions exerted by the EPNs and the invaded insects, respectively. Although EPN reproduction is possible only if their metabolism is shifted to enable infectivity (power to invade), reproductive capacity (yield per insect) and virulence (power to kill), the processes of EPN feeding and recycling comprise various morpho-histological, physiological and biochemical aspects [58–60]. Basic research is required to examine the genetic architecture of key traits, e.g., stress tolerance, infectivity and reproduction [61]. These aspects, as well as other molecular and structural demonstrations of the key events that are comprised in the insect–EPN interactions, should be accurately explored for better management of insect pests.

Briefly, the basic infection triangle—that insect mortality happens only when an insect host and its EPNs co-exist in a proper environment—shows facets that are focal to many approaches to expand the operations of these various interactions to the positive side of the nematodes. Within the wide concept of these interactions, all the above-mentioned aspects should be harnessed to control insect pests as best we can.

### 5.2. Current EPN Genetic Techniques to Optimize Insect–EPN Interactions

It is well-established that the genetic improvement of EPNs may comprise the screening of required traits in the natural population to discover the desirable trait(s), hybridization, selective breeding, mutagenesis and molecular genetics (recombinant DNA technology). As demanded traits are frequently based on polygenetical inheritance, a reliable technique for genetically boosting desired traits may be possible via screening and selective breeding [62,63]. An effective selective breeding project must possess the desired alleles in gene pools of the examined EPN populations with a relatively high heritability value. Interestingly, this was accomplished to enhance host finding [64,65], host pathogenicity [66,67], EPN efficacy [68], longevity [69], heat/cold tolerance [70,71], cost-effective application rate [72] and nematicide resistance [73]. On the contrary, deficient genetic variation in resistance to ultraviolet (UV) light led to the failure of such a breeding program to increase UV tolerance in *S. carpocapsae* [74]. Issues related to selective breeding are sometimes reflected in exclusive laboratory adaptation of the EPN strain especially when relaxed selection results in loss of field adaptation of the trait. Although a 72-fold increase in the host-finding ability of a particular *S. carpocapsae* strain against *Popillia japonica* (the Japanese beetle) was achieved in the laboratory breeding, no difference in infectivity was found relative to the wild-type strain under field conditions [64]. Notably, an efficacy gap between laboratory and field performance should be considered. This technique exerts an overall fitness cost to the selected strain too. To alleviate such defects, a selected strain

may be protected via cryopreservation in liquid N$_2$ [75] and the re-enforcement of selection pressure at even intervals [63,76]. In contrast, mutagenesis may be adequate where a few regulatory genes are responsible for the needed trait. It could be achieved in mutagenic strains with raised desiccation tolerance [77]. Ethyl methansulfonate (EMS) is famous for inducing mutagenesis. Yet, transposon mutagenesis is generally more reliable than chemical mutagenesis. The former has a higher frequency of mutations and lower lethal efficacy. It can also induce single-hit mutations, incorporate chosen markers in EPN strain construction and recover genes after mutagenesis [78]. The demerits of transposon mutagenesis include its low frequency in living systems and inaccuracy with most transposition systems. Mutant EPNs with obvious morphological characters can function as genetic markers for showing the desired trait(s) or for mapping of useful genes [79]. As novel EPN genes are detected, mutant rescue phenotype analysis can be a useful method for characterizing genes. Recently, mutagenesis screens could exploit the genetically tractable system of *S. hermaphroditum* to characterize its properties of reproduction [80]. This technique utilized this nematode species as a genetic development model to examine both insect parasitism and naturally occurring mutualistic symbiosis. Hybridization is also a robust technique used to combine beneficial traits, with good heritability in a natural population, to generate high-quality strains [81]. A heat-tolerant strain, *H. bacteriophora* ISS, could be hybridized with *H. bacteriophora* HP88, and heat tolerance was proved in the hybrid progeny by using a mutant-marker isolate *(Hp-dpy-2)* and backcrossing [70]. Moreover, fitness in the hybrid progeny was maintained relative to the parental strain [81].

Surely, combining more than one genetic improvement technique has evidenced powerful mechanisms for enhancing EPN performance. Sharifi-Far et al. [82] used a combination of the discovery of natural EPN populations and selection of superior inbred lines to enhance cold tolerance and the effectiveness of *H. bacteriophora* against *Delia radicwn* (the cabbage maggot). Selecting the most favorable lines from a homozygous group of inbred lines can be a robust method to develop EPN products with superior and stable useful traits [61,82].

Admittedly, genetic engineering provides a significant merit over the other techniques. It can generate small, defined advantageous changes in the EPN genotype. Hashrni et al. [83] reported the first successful transformation of heat-shock protein in *H. bacteriophora*, followed by Vellai et al. [84] for *S. feltiae*, both using the microinjection technique. Both used *Escherichia coli* Lac-Z-encoded enzyme 13-galactosidase (13-gal) as the reporter molecule for the transformation. Yet, because 13-gal needs a lengthy staining step, which severely affects the nematode, Hashmi et al. [85] suggested the jellyfish (*Aequorea vicloria*) green fluorescent protein (gfp) gene as a better selectable marker for gene expression in EPNs. In the widespread microinjection method, the transferred DNA may be ensured via three various manifestations, i.e., extrachromosomal array, integration into the genome or transient expression for only one or two generations [86]. These trials were almost the pioneers for EPN genetic improvements. Then, case studies of such methods were recently reviewed [61].

As genetic transformation allows for improvements that are not possible with classical methods for specific traits, various avenues are being developed [86]. For instance, introducing more copies of genes to back the already existing ones, e.g., the transformation of multiple endogenous cuticle-degrading proteases from entomopathogenic fungi [87]. Another avenue of providing superior genotype could be attained via adding new secreted bioactive compounds, like those encoded by a scorpion venom gene, into the genome of the EPF *Metarhizium anisopliae* [88]. As *H. bacteriophora* has relatively few predicted protease and protease inhibitors in its secretome, Baiocchi et al. [86] speculated that this small number may explain the heavy dependence of *H. bacteriophora* on its mutualistic bacteria for immune suppression of the insect host. If so, introducing more copies of genes responsible for EPN-secreted proteases can have more influence on nematode penetration into the insect haemolymph [89,90], immune suppression [91] and degradation of insect tissues [92]. Clearly, examples of transforming superior genes from fungi/bacteria

do not mean that the genetic constitution of definite EPN species/strains are devoid of different promising genes [89]. For instance, a unique heat stable, water-stress-related protein, designated DESC47 and encoded by the *desc-47* gene, was found in dehydrated the *S. feltiae* strain IS-6 [93]. Also, Wang and Gaugler [94] found EPN surface coat proteins in *S. glaseri* and recorded that at least one protein (SCP3a), when injected into an insect host, defends unrelated EPN species against encapsulation and latex beads against phagocytosis. Consequently, EPN strains/lines with superior biocontrol traits are being obtained via genetic improvements [59,62]. Nonetheless, genetic improvements to yield EPN strains with better traits should be continued in the future to develop upgraded EPNs so that they can be reliably and inexpensively used as bio-insecticides globally. Meanwhile, care must be taken so that the genetic improvement of some traits does not negatively affect other desirable traits, as it sometimes occurs [65].

### 5.3. Exploring EPNs Molecular Tools for Favorable Plant–Insect Interactions

Advanced control measures of insect pests are being developed to exploit molecular effectors in insect-immune modulation and pathogenicity by EPNs. For instance, both *S. carpocapsae* and *S. feltiae* IJs can deliver hundreds of proteins to their hosts; some of them are fatty acid- and retinol-binding proteins that act as potent modulators of the insects. This complex mixture of proteins can reduce host resistance to infection by EPN-mutualistic bacteria and contribute to host death [95]. Also, emerging evidence proposes that behavioral changes related to host finding may be regulated by neuropeptides. Neuropeptide characterization in *S. carpocapsae* was performed via sub-Dalton high-accuracy tandem mass spectrometry coupled with ultra-high performance liquid chromatography in order to know how EPNs regulate their behavior [96]. Such studies are essential to help boost EPN application and host specificity. Similar studies [95,97] aim at upgrading scientific assumptions for a better understanding and favorably directing insect–EPN interactions.

Therefore, conducting fundamental research for insect pest management via EPNs should continue as well. It is especially important because tools and devices used for advancing EPNs are still at a relatively early stage. Glazer and Shapiro-Ilan [61] focused on a few devices needed for the characterization of the heterogeneity or homogeneity of a population, such as sequence characterized amplified region (SCAR), random amplified polymorphic DNA (RAPD), restriction fragment length polymorphism (RFLP), amplified restriction fragment length polymorphism (AFLP), cleaved amplified polymorphic sequences (CAPSs), simple sequence repeats (SSRs), and single nucleotide polymorphisms (SNPs). These methods should be bound with elaborate classical and molecular approaches, such as transposon mutagenesis, expressed sequence tags (ESTs) screening, RNA interference (RNAi) technology, creating and rescuing mutant phenotypes, and marker-assisted selection (MAS), to bolster EPN reliability via cost-effective methods. For instance, advances are needed to develop markers to track transfer or (molecular, morphological and quantitative trait loci (QTL)) the enhancement/degradation of traits and to identify useful genes that can be transferred between EPN populations by molecular means.

### 5.4. Modern Examples of Progress Based on EPN-Molecular Techniques

Current methodologies with promising results are available for exploitation and expansion. Levy at al. [98] studied the osmotic desiccation tolerance among natural *H. bacteriophora* populations for genetic diversity. They exposed the IJs to polyethylene glycol 600. After the osmotic desiccation stress, genomic variation and gene expression patterns were examined. Thus, transcriptomic investigation enabled the characterization of genes and molecular markers as genetic selection tools to improve EPN tolerance to environmental extremes. To elevate the environmental stress resistance of EPNs as a priority for their biocontrol reliability, more than 80 *H. bacteriophora* wild-type (WT) strains and inbred lines were thoroughly characterized for their IJ longevity and EMS mutants, with extended survival have been bred through selection. Their phenotypic and genotypic information was subsequently combined to determine genes, DNA polymorphisms and genotypes

with high potential for further improvement [69]. The authors assumed consolidation of desirable EPN traits via incorporating high throughput genotyping screens for EPN breeding with subsequent promises of their application in the future. Also, field trials using standard farming machinery were conducted in different European countries to control the Western corn rootworm (*Diabrotica virgifera virgifera*) by commercial *H. bacteriophora*. It was applied at $2 \times 10^9$ ha$^{-1}$ with 200 L water ha$^{-1}$ into the furrow together with the maize seeds. The nematodes caused a 65% reduction in the pest population level and surpassed results gained by the chemical standards in 11 of the 16 trails [72]. A notable example used a more virulent and persistent *H. bacteriophora* line from genetic breeding, which could justify a reduction in the EPN application rate to one-half that of the commercial nematodes. Consequently, it has the merit of decreasing costs to the range of synthetic chemicals [72].

Ehlers et al. [99] have recently evidenced attributes of *H. bacteriophora* as a super model for genetic amelioration in the biocontrol of insect pests. Hence, emerging technologies should be adapted for EPN genetic application to incorporate new genomic, bioinformatics and proteomic knowledge. As the *H. bacteriophora* (TT01 strain) genome has already been sequenced [89], novel devices may be used for genome editing programs to develop super strains. Our perception is that such programs for genetic improvement should be expanded with more EPN products to prove their merits but under natural/field conditions. The clustered, regularly interspaced short palindromic repeat (CRISPR)/CRISPR-associated protein (CRISPR/Cas), as a robust tool for precisely targeted gene modification, may be used to generate variation and hasten breeding schemes [100]. The effective use of CRISPR/Cas9-directed genome editing may aid in gaining desired traits with a limited genetic pool. Ibrahim et al. [101] recorded four major methods of gene editing in order to back general breeding for resistance: recombinase-mediated site-specific gene integration, homologous recombination-dependent gene targeting, oligonucleotide-directed mutagenesis and nuclease-mediated site-specific genome modifications. Eventually, this review will handle how to induce favorable plant–insect interactions via briefing on a few main factors related to production practices in the three basic biocontrol methods used by EPNs. To conclude the main topic, suffice to say, some examples vividly express their counterparts.

## 6. Biocontrol Methods of Insect Pests by EPNs

### 6.1. Augmentation or Inundative Biocontrol

Because the most common use of EPNs is for the augmentation tactic, related biotic and abiotic actors that affect EPN biocontrol efficacy were recently reviewed [4,33]. Given the shortest time among the basic biocontrol methods that this tactic goes through, these factors could be relatively harnessed. In this tactic, the nematodes are applied inundatively as BCAs to rapidly kill the pest within a few days. However, the tactic requires not only choosing the best EPN via host matching but also agricultural operation and timing adequate for EPN survival and efficacy. Jaffuel et al. [102] recorded trivial pest control when rootworms attacking maize began feeding a week prior to EPN application. On the contrary, they reported that the well-timed usage of alginate beads that encapsulate the EPNs brought about a high reduction ($p < 0.05$) in the banded cumber beetle *Diabrotica balteata*-inflicted root damage relative to that of EPNs in water suspension. Furthermore, although *S. glaseri* is quite effective against *Agrotis ipsilon* [102,103], the insect mortality via baiting in plots with IJ inoculation soon after peanut seeding was mostly higher than that in plots with later IJ inoculation. The higher EPN efficacy was likely because of the non-contact of IJs with the chemical fertilizers for 7 days; EPNs added before these chemicals showed better efficacy than EPNs that were tank-mixed with chemicals [54]. Consequently, plots treated with *S. glaseri* that were not tank-mixed with chemicals simultaneously lead to high peanut germination rates close to those of the chemical insecticide chlorpyrifos. The authors reported that inorganic phosphorus fertilizers had more adverse effects than others on *S. glaseri* virulence and subsequent peanut germination than potassium and nitrogenous chemical fertilizers. Therefore, agricultural practices that can increase [54,102] or diminish [104] EPN biocontrol of host pests should be considered.

### 6.2. Classical Biocontrol

In contrast to the common augmentation tactic, classical biocontrol using EPNs stems from a directed search of settings in which the EPNs will have to develop the demanded attribute. *Steinernema scapterisci* imported from Uruguay is still used as classical biocontrol agent against invasive mole crickets affecting pastures and turf in Florida (USA). The species was isolated from Uruguay as one of the assumed centers of origin of the mole crickets [105]. Nonetheless, relatively cheap chemical insecticides of turf have minimized the use of the nematode products to control mole crickets [106]. Due to many EPN surveys (discovery) as well as commercial and genetic improvements, most marketed nematodes are not currently used against their original host, which limited the classical biocontrol.

### 6.3. Conservation Biocontrol

To increase the persistence and efficacy of EPNs, conservation biocontrol aims to combine production practices favorable for EPN activity while limiting those exercises that damage them. In other words, it targets the modification of the ecological settings and production practices to maintain and enhance EPN effectiveness in order to decrease losses from insect pests. This strategy to biocontrol pests/pathogens has not received enough attention compared with the aforementioned inundative applications of EPNs. Yet, conservation biocontrol mainly considers the importance of unraveling the complexities associated with biotic/abiotic factors that affect the long-term performance of EPNs. It addresses the production practices that modulate EPN activity/level within the soil food web [107]. For instance, edaphic factors can offer insights on modulating soil features to boost biocontrol by favoring/maintaining certain settings. Mulching [108], soil texture and moisture [109], salinity [110] and pH [111] could modulate EPN population levels directly/indirectly by impacting their hosts/enemies [111]. Although changes in the environment to create favorable conditions for EPN persistence and efficacy are required, the cost/benefit of the related practices needs to be evaluated carefully as it differs from small vs. large farms.

Although records of pest management consider the root weevil (*Diaprepes abbreviatus*) in Florida citrus groves as an outstanding biocontrol gaining, EPN efficacy there relies heavily on soil properties [112]. However, as EPNs are used inundatively there to rapidly kill the pest, little interest was initially given to the EPNs' superior efficacy in the well-draining coarse sands of the Central Ridge relative to their less effective biocontrol in the poorly draining fine-textured soils of the Flatwoods. Thereafter, the substitution of sand for native soil was an effective way of conserving EPNs and demonstrated promise as a cultural practice to control *D. abbreviatus* in flatwoods groves with significant weevil damage to citrus trees. Modifying soil there is a striking example of conserving EPN efficacy, decreasing weevil herbivory, and enhancing tree growth and fruit yield [107]. While these groves depended on variables associated with the water content of soil for the abundance and diversity of EPN species, such variables also modulated the soil food web assemblage. Thereafter, shifting agricultural practices to fight against the accidental introduction of huanglongbing disease altered the soil properties and food web structure in ways that minimized the conservation biocontrol potential by EPNs [109].

Recent assessment of EPNs and related biotic factors via classical and molecular tools, combined with the impact of edaphic factors in various agroecosystems, may offer further conservation biocontrol perceptions. Their promise will rely on the specific scenario. Obviously, annual crops subjected to alterations from one season to another without a stable rhizosphere are frequently less effective for conservation biocontrol by EPNs. In Switzerland, the natural occurrence of EPNs in various long-term field trials, including annual crops such as wheat and maize, was low relative to their occurrence in natural areas. In these scenarios, the low EPNs numbers and high numbers of other microorganisms implied that EPNs are unsatisfactorily controlling root-feeding pests of such annual crops [104]. Perennials, such as citrus, mango and guava trees, have stable soil environments and are more conducive to such conservation biocontrol. In other settings, shaping the ecological

structures might assist in backing the EPN activity, but with a distinguished impact relying on the growing plants. For instance, while the cover crops did not enhance the occurrence of natural EPNs in annual Swiss crops, spontaneous cover crops planted within Spanish vineyards lines favored EPN occurrence relative to classical tillage practice [113]. In another horticulture crop, the habitats of citrus trees in reclaimed desert favored EPN abundance relative to old farmlands of the Nile Basin [114]. Yet, superior traits of indigenous EPN strains/isolates against specific pest(s) [115] may be utilized to potentiate conservation biocontrol.

Eventually, to materialize conservation biological control with EPNs as a reality, it is imperative to better understand the multitrophic interactions, mechanisms of the used BCA, and edaphic/biotic factors affecting its performance. This necessitates solving the complexities relevant to the best habitat adjustments that back EPN survival and activity of their mutualistic bacteria under field conditions.

## 7. Avoiding Unfavorable Aspects of EPN for Further IPM Exploitation

Having referred to impediments of EPN price and reliability, favorable approaches should be substantially deemed to optimize EPN uses. Vivid examples to face such obstacles (Table 1) should be put into current/emerging strategies to develop and expand EPN commercial usage. Admittedly, EPNs are applied with notable success in various continents (Table 2). Yet, reduced EPN application density via virulent and persistent nematodes could bring the costs into the range of chemical pesticides [72]. Notwithstanding that most references focus on the use of EPNs or their symbionts against individual species of plant pests, there are frequently more than one susceptible pest/pathogen as targets in the same field. Approaches for collective control of several susceptible pests/pathogens simultaneously should be earnestly attempted. On the other hand, gene editing for better EPN performance must be wisely utilized to avoid undesired effects, e.g., pleiotropic impacts on qualitative and quantitative traits [65]. Consequently, this review proposes to apply the influences of these methods on a case-by-case basis in order to follow and improve desired/specific traits according to the variables that dictate EPN performance. It will help in fixing the exact factors controlling the gene expressions and combining them with other control tactics into IPM. Like other BCAs [116], common factors negatively affecting success of EPNs should be carefully avoided. While biological/ecological factors should be harnessed to serve their biocontrol activities, many metabolic syntheses of EPNs/mutualistic bacteria [14,15,86] with their useful functions should be further explored. This will enable us to understand their modes of action and perfect their usage in IPM.

**Table 1.** Examples of various strategies for boosting entomopathogenic nematode (EPN) and symbiont uses against plant pests.

| Strategy | EPN/Bacterial Species | Target Pests or Media Used | Target Beneficial Traits/Objectives | References |
|---|---|---|---|---|
| 1. Genetic improvement | | | | |
| (A) Discovery of new species or strains | *Steinernema scapterisci* | Mole crickets (*Scapteriscus* spp.) | Efficacy against invasive mole crickets inflicting pasture and turf | [105] |
| (B) Selection/breeding of promising EPNs | | | | |
| i. Enhancing foraging strategy | *S. carpocapsae* and *Heterorhabditis bacteriophora* | Red palm weevil (*Rhynchophorus ferrugineus*) | High host-seeking ability | [65] |
| ii. Raising drought tolerance | *H. megidis* and *H. bacteriophora* | Greater wax moth (*Galleria mellonella*) | Enhanced desiccation tolerance | [77,117] |
| iii. Boosting tolerance to temperature extremes | *H. bacteriophora* | Greater wax moth (*Galleria mellonella*) | Tolerance to temperature extremes | [118] |

**Table 1.** *Cont.*

| Strategy | EPN/Bacterial Species | Target Pests or Media Used | Target Beneficial Traits/Objectives | References |
|---|---|---|---|---|
| iv. Securing EPN virulence under UV-stressed conditions | *S. carpocapsae* and *S. riobrave* | Greater wax moth (*Galleria mellonella*) | Keep the virulence of UV-stressed nematodes in warm/cold ambient | [119] |
| v. Boosting EPN virulence | *S. feltiae* | Western flower thrips (*Frankliniella occidentalis*) | Increased infectivity and efficacy | [120] |
| vi. Increasing nematicide resistance | *H. bacteriophora* strain HP88 | Greater wax moth (*Galleria mellonella*) | Improve resistance to fenamiphos, oxamyl and avermectin | [73] |
| vii. Breeding EPN for cost-effective application | *H. bacteriophora* | Western Corn Rootworm (*Diabrotica virgifera virgifera*) | Reducing EPN application density to bring costs into the range of chemical pesticides | [72] |
| viii. Extending the survival | *H. bacteriophora* | Greater wax moth (*Galleria mellonella*) | Improving stress tolerance and survival | [69] |
| (A) Nematode-genetic engineering | | | | |
| i. Raising thermotolerance | *H. bacteriophora* | Turfgrass field microplots | Heat tolerance via transforming a heat shock protein | [121] |
| ii. Raising osmotolerance and desiccation tolerance in the transgenic adults | *S. feltiae* | Laboratory bioassays | Improve osmotolerance and desiccation tolerance in the modified EPN adults | [84] |
| (B) Plant-incorporated protectants | | | | |
| i. Enhancing plant tolerance against aphids | *Xenorhabdus bovienii* | Peach-potato aphid (*Myzus persicae*) | Expressing protease inhibitor protein to enhance insect tolerance | [122] |
| ii. Bacterial mixture to control an insect pest | *P. temperate temperata* culture broth | Brassica leaf beetle (*Phaedon brassicae*) | The bacterial cultured broth showed potent immunosuppressive activity | [21] |
| 2. Non-genetic improvement | | | | |
| i. Raising EPN yield and fitness | *S. feltiae* SN strain | Optimized in vitro solid culture media | Improve EPN yield and fitness against *Spodoptera litura* | [123] |
| ii. Inducing high EPN recovery and yield | *S. jeffreyense* and *S. yirgalemense* | Optimized in vitro liquid culture medium | Obtaining high EPN recovery and yield | [124] |
| iii. Improved formulation | *S. carpocapsae* | Lesser peachtree borer (*Synanthedon pictipes*) | Enhanced IJ survival and induced high pest mortality | [125] |
| iv. Dual-purpose: | *H. bacteriophora* strain EGG | Both insect and plant parasitic nematode pests | Multi-purpose usage of EPNs | [30] |
| v. Co-application: | Formulate *S. carpoca- psae* with *S. feltiae* | Black vine weevil (*Otiorhynchus sulcatus*) larvae | Keep the pest populations below the economic threshold level | [48] |
| vi. Sequential application of *Metarhizium anisopliae* at 0, 7, or 14 days prior to EPN | *H. bacteriophora*, *S. carpocapsae* or *S. kraussei* | Black vine weevil (*Otiorhynchus sulcatus*) larvae | Synergistic or additive effect of the fungus and an EPN species | [35] |

**Table 2.** Examples of EPN species, insects targets and efficacy in various continents under field conditions.

| Country (Continent) | Crop | Insect Target | EPNs | Mortality | References |
|---|---|---|---|---|---|
| USA (North America) | Citrus | *Diaprepes abbreviatus* | *Steinernema riobrave* | 77–90% | [126] |
| Italy (Europe) | Nature Parks | *Popillia japonica* | *Heterorhabditis bacteriophora* | 44–93% | [127] |
| Colombia (South America) | Banana | *Metamasius hem-ipterus sericeus* | *Steinernema colombiense* | Adult (50%) Larvae (90%) | [128] |
| Brazil (South America) | Sugarcane | *Mahanarva fimbriolata* | *Heterorhabditis* sp. | 74% | [129] |
| Egypt (Africa) | Peanut | *Agrotis ipsilon* | *S. glaseri* | 85.8–93.3% | [54] |
| China (Asia) | Chinese chive | *Bradysia odoriphaga* | *Heterorhabditis* sp. & *S. bibionis* | 69%% | [130] |
| Australia | Banana | *Cosmopolites sordidus* | *Steinemema carpocapsae* | Up to 68% of infected larvae | [131] |

## 8. Conclusions

Developing cost-effective EPN applications as reliable bio-pesticides especially for IPM schemes of plant-insect pests are being addressed as one of the present pressing issues. The current research is mainly focused on commercial and genetic improvements, e.g., enhanced strain efficacy and tolerance to environmental settings, cost-effective mass production, improved formulation and application technology, combinations with other cultural inputs and fitting into more edaphic and biological environments. Novel trends for the better exploitation of EPNs should widen the area of their activity spectra. The techniques, tools and knowledge developed for genetic improvement of EPNs along with proper agricultural exercises should be a way forward in plant protection and managing insect pests. Combining production practices with genetic improvement schemes via sophisticated molecular and non-molecular methods is the best approach for transformational technology to upgrade EPNs' biopesticidal position. Notwithstanding the current focus on using EPNs or their symbionts against individual species of insect pests, there is frequently more than one plant pest/pathogen susceptible to infection by these BCAs in the same field. This could result in an underestimation of their application value. Therefore, inclusive improvement plans that comprise diverse beneficial biocontrol traits are still required. A good arsenal of such traits will enable us to optimize EPN performance based on the given variables.

**Funding:** This research was funded by STDF, US-Egyptian Research Program of project "Preparing and evaluating IPM tactics for increasing strawberry and citrus production" cycle 17 grant number 172 and the NRC in-house project No. 13050112 entitled "Pesticide alternatives against soil-borne pathogens and pests attacking economically significant export crops" carried out by the National Research Center, Egypt.

**Institutional Review Board Statement:** Not applicable.

**Informed Consent Statement:** Not applicable.

**Data Availability Statement:** Not applicable.

**Acknowledgments:** This article is supported in part by the US-Egypt Project cycle 17 (no. 172) entitled "Preparing and evaluating IPM tactics for increasing strawberry and citrus production". The study was also supported in part by the NRC In-house Project No. 13050112 entitled "Pesticide alternatives against soil-borne pathogens and pests attacking economically significant export crops" carried out by National Research Center, Egypt. The author thanks Zafar Handoo of the USDA, USA, for his valuable comments on the manuscript.

**Conflicts of Interest:** The author declares no conflict of interest. The funders had no role in the design of the study; in the collection, analyses, or interpretation of data; in the writing of the manuscript, or in the decision to publish the results.

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
