# Peer review of "Optimizing Entomopathogenic Nematode Genetics and Applications for the Integrated Management of Horticultural Pests"

_horticulturae, doi:10.3390/horticulturae9080865_

Round 1

Reviewer 1 Report

Overall, this manuscript is addressing an important topic related with the use and improvement of Entomopathogenic Nematodes.  I suggest to add three aspects that may improve the scientific quality of this manuscript. 

See file attached. 

Author Response

I am deeply thankful for all your helpful suggestions to improve this study; I have incorporated them in this new version of the manuscript as described below and in the “Supplementary manuscript” attached. I tried to comply with all the comments and suggestions as best I can as follow: COMMENT: Overall, this manuscript is addressing an important topic related with the use and improvement of Entomopathogenic Nematodes. I suggest to add three aspects that may improve the scientific quality of this manuscript:
1-Some photographs about EPNs: (Heterorhabiditis spp. and Steinernema spp.). Some target insect pests infected by EPNs and an example of commercially available EPNs. Those photos can illustrate much better the topic and made the reading more pleasant.
Answer: Thanks for the helpful suggestions to improve this study. I have incorporated 3 photos as parts of Figure 1 in this new version of the manuscript. Also. the following sentences were added as examples of commercially available EPNs: Recently, commercial development of at least five Heterorhabditis species and eight Steinernema species were reported [4]. Examples are H. bacteriophora, H. indica, S. carpocapsae, S. feltiae, and S. riobrave……… Examples are presented (Figure 1) to show these EPNsymbiont complexes killing and multiplying in their insect hosts and then leaving them to search for other insect hosts…… Figure 1. Dead larvae of cabbage butterfly, Pieris brassicae A) killed by Steinernema sp., B) killed by Heterorhabditis sp., and C) emergence of H. bacteriophora after parasitizing an adult of the strawberry sap beetle
Lobiopa insularis [33].

COMMENT: 2-Commercial production of EPNs. A short paragraph addressing the
current rearing system of EPNs with focus in aspects that need to be improve. One of the reasons by EPNs are not used more broadly is due to the high cost of production and their variable efficacy under field conditions. Improvements in the rearing systems could be low the costs.
Answer:The following was added: Also, a thorough analysis of EPN production methods with their merits/demerits for relevant exploitation and favorable formulations were recently summarized [56]. Three production methods have significantly contributed in the introduction of
EPNs to the marketplace globally. Fine-tuning of these methods to reduce costs and increase EPN virulence should continue to contribute to EPNs expanded applications. In vitro liquid massproduction is the best economical method and is likely ongoing to prevail the quantity of EPNglobal production. On the other hand, in vitro solid production may be still competitive especially where labor is less costly. Although in vivo production is the least economical method, it will
likely go ahead to be adequate for certain niche markets or for certain small or start-up firms. Further optimization of in vivo production [30-32] may boost cost efficiency. Meanwhile, it is likely that the economy of scale is reliable for boosting in vitro approaches including both methods of solid and liquid media [56]. For example, pheromone extracts that act as boosters for EPN efficacy was reported [57] and may be promising components in the media of these in vitro
approaches.

COMMENT: 3-A table showing examples of EPNs species, insects target and efficacy under field conditions. We already know that there is a huge gap between EPNs efficacy under laboratory conditions and field conditions. So, all molecular techniques that you are discussion for the improvement of EPNs efficacy are justified.
Answer: Sincere thanks. Your suggestion was considered and a new table was inserted to give examples of EPN species, insects targets and efficacy in various continents (not only one continent) under field conditions.

COMMENT: In section 6.2 Conservation Biocontrol. Try to do changes in the
environment for facilitate conditions to EPNS survey is okay, always and when an
economic benefit is obtained. In a small farm with focus on eco-friendly crop production or organic production, maybe those recommendations are useful and maybe practical to do. However, in a big operation with hundreds of hectares try to do those modifications may be expensive and are not practical. The cost/benefit of the practices need to be evaluated carefully.
Answer: OK. I added: Although changes in the environment to create favorable conditions for EPN persistence and efficacy is required, the cost/benefit of the related practices needs to be evaluated carefully as it differs from small versus large farms.
COMMENT: Overall, in some part of the manuscript the reading is difficult to
understand, very long paragraphs that made the reading boring. Use of short paragraphs with specify examples is much better. Finally, there are some long statement with factual information that have no references. Example: Section 2. ENP Biology and Ecology: The EPN invasion into an arthropod host may …….. (there is 14 lines and not references about this specific description of ENPs biology).
Answer: OK. The paragraph was modified and references were inserted in the
aforementioned 14 lines.

Reviewer 2 Report

Abstract Section

Major comments:

The paper offers a comprehensive review on entomopathogenic nematodes (EPNs), focusing on their biology, ecology, and their relationship with symbiotic bacteria for pest management. It further discusses the application of various genetic techniques for improving EPN efficacy and tolerability under various settings. However, the transitions between these focuses could be improved for a smoother reading experience. For example, 

§  The transition between EPN biology/ecology and the discussion of EPN-bacteria symbiosis: The abstract transitions abruptly from an overview of EPN biology to the topic of symbiotic bacteria. A transition sentence linking these concepts might improve the flow.  ‘Understanding the biology and ecology of EPNs, particularly their symbiotic relationships with bacteria, is crucial to their effective use in pest management.’ 

§  The transition from EPN-bacteria symbiosis to their role in IPM. After discussing the EPN-bacteria symbiosis, the abstract abruptly moves to their roles in IPM. A smoother transition could emphasize why this symbiotic relationship is relevant to IPM, e.g., ‘This symbiotic relationship between EPNs and bacteria plays a key role in IPM, providing unique advantages.’ 

§  Transition to genetic improvement techniques. From the discussion on IPM, although the author mentioned that ‘to supplant less expensive and more effective but toxic/unhealthy pesticides.’, the abstract still appears to jump to the genetic improvement of EPNs. Here, the relationship between IPM efficacy and the potential genetic enhancement of EPNs could be established more clearly, e.g., ‘To maximize the effectiveness of EPNs in IPM, various genetic improvement techniques are being explored.’

Minor comments:

§  The term ‘transformational technology’ is somewhat ambiguous and could benefit from a clear definition. Is it referring to genetic engineering, new application methods, or other biotechnological advancements? An improved explanation will guide the reader’s understanding.

§  The phrase ‘other merits of EPNs’ is quite broad. Could this be replaced with specific examples of these merits or a brief overview of what they encompass? It would provide more information to the reader upfront about what the paper covers before going through the main content.

Main content

Major comments:

§  After examining the main content in this manuscript and the main content in the same author’s another publication in Egyptian Journal of Agronematology (EJAN) (https://doi.org/10.21608/ejaj.2023.280551), the reviewer can say that these two review papers are not identical, but they do share a similar thematic focus, terminology, and structure. They both focus on the topic of EPNs, their discovery, and their role as biological control agents. For example, 

o   Thematic focus: Both introductions focus on the same overall theme: the role and application of Entomopathogenic nematodes (EPNs) as biological control agents (BCAs). 

In horticulturae submission: "Entomopathogenic nematodes (EPNs)... form a notable ground for their practical and further use as biocontrol agents (BCAs) of many insect pests."

In EJAN: "Entomopathogenic nematodes (EPNs)... formed a foundation of pioneering related research that has resulted in major advances in biological control."

o   Structure: In the Introduction section, both manuscripts follow a similar overall structure, beginning with a broad introduction to EPNs and their roles as BCAs, followed by a discussion of their bacterial symbionts, and concluding with a discussion of the challenges and future directions in the field.

In horticulturae: Starts with a general overview of EPNs and their bacterial symbionts, then discusses the role of the bacteria in biocontrol, challenges and future directions in the field.

In EJAN: Follows a similar pattern, introducing EPNs and their symbionts, then detailing their evolution into BCAs, and finishing with challenges and future directions in the field.

§  Also, the reviewer notice that there seems specific content overlap, but no more advanced research reports mentioned in the horticulturae submission, compared with the author’s 2023 publication in EJAN. For example,

o   In horticulturae submission: it's mentioned "the species of the EPN and their mutualistic bacteria are being elevated in their numbers. Currently, about 22 Heterorhabditis and 102 Steinernema species have been recorded [11-13]." 

o   In EJAN publication: it states "the numbers of the EPN species and their mutualistic bacteria are always subject to increase. At least 102 Steinernema and 22 Heterorhabditis species have been identified to date".

§  Both papers mention the foraging strategies of the EPNs. 

o   In horticulturae submission: "The only free living EPN stage is the specialized third stage juvenile, called the infective juvenile (IJ) or dauer juvenile. It can actively seek and invade the host insect. Foraging strategies of these juveniles have two distinct ends of ambusher-cruiser continuum." 

o   In EJAN publication: it explains similarly: "A specialized third stage juvenile, usually called the dauer juvenile or infective juvenile (IJ) is the only free living stage that can actively invade the insect host. Foraging strategies of IJs have two different extremes within the foraging continuum; ambushers versus cruisers."

Maybe the reviewer missed something; however, based on their current understanding, the reviewer cannot definitively ascertain whether this review submission adequately presents concise and precise updates on the latest progress made in optimizing EPN genetics and applications for IPM or is of interest to the scientific community.

 Minor editing of English language is required.

Author Response

I am deeply thankful for all your helpful suggestions to improve this study; I have incorporated them in this new version of the manuscript as described below and in the “Supplementary manuscript” attached. I tried to comply with all the comments and suggestions as best I can as follow:
COMMENT: Abstract Section: Major comments: The paper offers a comprehensive review on entomopathogenic nematodes (EPNs), focusing on their biology, ecology, and their relationship with symbiotic bacteria for pest management. It further discusses the application of various genetic techniques for improving EPN efficacy and tolerability under various settings. However, the transitions between these focuses could be improved for a smoother reading experience. For example: § The transition between EPN
biology/ecology and the discussion of EPN-bacteria symbiosis: The abstract transitions abruptly from an overview of EPN biology to the topic of symbiotic bacteria. A transition sentence linking these concepts might improve the flow. „Understanding the biology and ecology of EPNs, particularly their symbiotic relationships with bacteria, is crucial to their effective use in pest management.‟
§ The transition from EPN-bacteria symbiosis to their role in IPM. After discussing the EPN-bacteria symbiosis, the abstract abruptly moves to their roles in IPM. A smoother transition could emphasize why this symbiotic relationship is relevant to IPM, e.g., „This symbiotic relationship between EPNs and bacteria plays a key role in IPM, providing unique advantages.‟
§ Transition to genetic improvement techniques. From the discussion on IPM, although the author mentioned that „to supplant less expensive and more effective but toxic/unhealthy pesticides.‟, the abstract still appears to jump to the genetic improvement of EPNs. Here, the relationship between IPM efficacy and the potential genetic enhancement of EPNs could be established more clearly, e.g., „To maximize the effectiveness of EPNs in IPM, various genetic improvement techniques are being explored.‟
Answer: OK. The aforementioned three transitions were complied with, accordingly your statements were inserted in the abstract.
COMMENT: Minor comments: § The term „transformational technology‟ is
somewhat ambiguous and could benefit from a clear definition. Is it referring to
genetic engineering, new application methods, or other biotechnological
advancements? An improved explanation will guide the reader‟s understanding.
Answer: OK. For more carifications, sentences were modified to be as follows: Yet, EPN cost and efficacy need transformational technology to supplant less expensive and more effective but toxic/unhealthy pesticides. Such a technology that fulfills significant uptake of commercial EPNs should be based on both boosting their market suitability and genetic improvement.

COMMENT: Minor comments: § The phrase „other merits of EPNs‟ is quite broad. Could this be replaced with specific examples of these merits or a brief overview of what they encompass? It would provide more information to the reader upfront about what the paper covers before going through the main content.
Answer: OK. For more clarifications, sentences were modified to be as follows:
Merits such as rapid killing of insect pests, ease of EPN/the symbiont mass
production and broad host range are presented in order to widely disseminate
conditions under which the EPN usage can offer a cost-effective or/and value-added techniques to IPM. Consequently, in the main text, more details were given such as three photos of infected insect pests and emergence of EPN juveniles in huge numbers from the infected insect, a paragraph on EPN mass production, and a table presenting examples of effective EPN control against insect hosts from various continents are added (Note: this was also requested by the other reviewer).
COMMENT: Main content
Major comments:
§ After examining the main content in this manuscript and the main content in the same author‟s another publication in Egyptian Journal of Agronematology (EJAN) (https://doi.org/10.21608/ejaj.2023.280551), the reviewer can say that these two review papers are not identical, but they do share a similar thematic focus, terminology, and structure. They both focus on the topic of EPNs, their discovery, and their role as biological control agents. For example,
o Thematic focus: Both introductions focus on the same overall theme: the role and application of Entomopathogenic nematodes (EPNs) as biological control agents(BCAs). In horticulturae submission: "Entomopathogenic nematodes (EPNs)... form a notable ground for their practical and further use as biocontrol agents (BCAs) of many insect
pests." In EJAN: "Entomopathogenic nematodes (EPNs)... formed a foundation of pioneering related research that has resulted in major advances in biological control."
o Structure: In the Introduction section, both manuscripts follow a similar overall
structure, beginning with a broad introduction to EPNs and their roles as BCAs,
followed by a discussion of their bacterial symbionts, and concluding with a
discussion of the challenges and future directions in the field.
In horticulturae: Starts with a general overview of EPNs and their bacterial
symbionts, then discusses the role of the bacteria in biocontrol, challenges and future directions in the field.
In EJAN: Follows a similar pattern, introducing EPNs and their symbionts, then detailing their evolution into BCAs, and finishing with challenges and future
directions in the field.
§ Also, the reviewer notice that there seems specific content overlap, but no more advanced research reports mentioned in the horticulturae submission, compared with the author‟s 2023 publication in EJAN. For example,
o In horticulturae submission: it's mentioned "the species of the EPN and their
mutualistic bacteria are being elevated in their numbers. Currently, about 22
Heterorhabditis and 102 Steinernema species have been recorded [11-13]."
o In EJAN publication: it states "the numbers of the EPN species and their
mutualistic bacteria are always subject to increase. At least 102 Steinernema and 22 Heterorhabditis species have been identified to date".
§ Both papers mention the foraging strategies of the EPNs.
o In horticulturae submission: "The only free living EPN stage is the specialized
third stage juvenile, called the infective juvenile (IJ) or dauer juvenile. It can actively seek and invade the host insect. Foraging strategies of these juveniles have two distinct ends of ambusher-cruiser continuum."
o In EJAN publication: it explains similarly: "A specialized third stage juvenile,
usually called the dauer juvenile or infective juvenile (IJ) is the only free living stage that can actively invade the insect host. Foraging strategies of IJs have two different extremes within the foraging continuum; ambushers versus cruisers."
Answer: OK. As the reviewer said that these two review papers are not identical, but they do share a similar thematic focus, terminology, and structure; I‟d like to add that the main objective of the two papers are quite different. The one published at EJAN aims to expand EPN applications especially to pests of livestock and honey bees. Therefore, it contained different main titles and subtitles such as:
Main title: Entomopathogenic Nematodes against Pests of Farm Animals
Subtitles:
a) Problems facing the application of pest control
b) Pests and diseases of farm animals compared to plant pests in term of EPN
application
Main title: The Potential of EPN-Based Biopesticides for Control of Honey Bee Pests
Subtitles:
a) The importance of Honey bees for human wellbeing and the so called “forgotten economy”
b) The greater wax moth Galleria mellonella and its impact on bee products and bee colony
c) Small hive beetle (SHB) Aethina tumida
Of course, addressing these titles and subtitles are NOT even related to the overall theme of
this journal “Horticulturae”. Yet, as you know fundamentals/basics of EPNs such as biology, ecology, efficacy...etc. of EPNs should be stated and would overlap with other REFs not only the one published at EJAN. The manuscript in its revised form adequately presents concise and precise updates on the latest progress made in optimizing EPN
genetics and applications for IPM, it addresses references published in year 2023 as well as other recent years of 2021 and 2022. These are much appreciated comments that I tried as best I can to comply with.

Round 2

Reviewer 1 Report

Now the manuscript looks much better. 

Authors have improved this manuscript in aspects that  were recommended by reviewers. 

Good job!

Reviewer 2 Report

The authors have made significant revisions to the manuscript, which are appreciated. The responses to the reviewer's comments were clear and addressed the majority of the concerns.

 Minor editing of English language might be required.